# The Impact of Stress from Social Isolation during the COVID-19 Pandemic on Psychiatric Disorders: An Analysis from the Scientific Literature

**DOI:** 10.3390/brainsci13101414

**Published:** 2023-10-05

**Authors:** Amanda Gollo Bertollo, Geórgia de Carvalho Braga, Paula Teresinha Tonin, Adriana Remião Luzardo, Margarete Dulce Bagatini, Zuleide Maria Ignácio

**Affiliations:** 1Laboratory of Physiology Pharmacology and Psychopathology, Graduate Program in Biomedical Sciences, Federal University of Fronteira Sul, Chapecó 89815-899, SC, Brazil; amandagollo@gmail.com; 2Cell Culture Laboratory, Graduate Program in Biomedical Sciences, Federal University of Fronteira Sul, Chapecó 89815-899, SC, Brazil; braga.georgia18@gmail.com (G.d.C.B.); margaretebagatini@gmail.com (M.D.B.); 3Nursing Department, University Center of Maringá, Maringá 87080-000, PR, Brazil; toninpaula@yahoo.com.br; 4Research Laboratory in Health Management, Innovation and Technologies, Graduate Program in Nursing, Federal University of Fronteira Sul, Chapecó 89815-899, SC, Brazil; luzardoar@gmail.com

**Keywords:** COVID-19, social isolation stress, neuropsychiatric disorders, biopsychosocial vulnerability, therapeutic strategies

## Abstract

The COVID-19 pandemic generated, in addition to severe symptoms, hospitalizations and deaths worldwide, as well as stress from the fear of the disease and social uncertainties, from restriction measures and social isolation. Stress from social isolation impacts mental health, aggravating existing conditions and triggering neuropsychiatric symptoms in individuals with biopsychosocial vulnerability. During and immediately after the period of social restriction imposed by the pandemic, the scientific community carried out several research protocols. These revealed results that relevantly demonstrate the harmful effect of the stress induced by the pandemic situation. This narrative review reports and discusses research results demonstrating impairments in psychiatric disorders such as autism spectrum disorder, dementia, eating disorders, schizophrenia, anxiety, and depression. In this sense, the community has identified a significant negative influence of social isolation on the mental health of individuals through the modification of individual routines and the absence of social interactions. Moreover, the community identified perceived differences related to the impacts on men and women. In addition to studies showing the effect of social isolation on disorders, an evaluation of protocols with some possible therapeutic intervention strategies during times of social restriction was developed.

## 1. Introduction

The COVID-19 pandemic resulted in thousands of hospitalizations and deaths, causing instability in global social, political, and health systems. However, the impacts of this pandemic were not limited to deaths and hospitalizations; new health issues were raised based on the need for quarantine [1]. The social isolation imposed by the quarantine led to the increase and worsening of multiple physical and mental disorders, which were sometimes capable of interacting with each other or impacting pre-existing diseases, thus putting the health of countless individuals at risk [2].

According to the World Health Organization (WHO), health encompasses the individual’s physical, mental, and social well-being [3]. In this sense, it is known that social isolation can negatively impact physical and psychological well-being and therefore health in general [4]. During the COVID-19 pandemic, this was no different. Social isolation impacted the health conditions of many groups, from teenagers to the elderly [2,5]. Moreover, data from different countries have shown that despite the differences in the manifestations caused by social isolation due to the altered management of the pandemic by each government and income level, the impacts of social isolation had worldwide repercussions, being more or less severe in different countries according to their different characteristics [6,7].

In addition to the already known effects caused by SARS-CoV-2, social isolation was able to generate serious consequences for world populations due to the stress it generates and its implications in most different systems [8]. A study by Socrates and colleagues noted a genetic correlation between social isolation and autism spectrum disorder, schizophrenia, and depression [9]. Accordingly, studies indicate that some of the changes that were very evident during the pandemic were those related to brain functions and psychological processes, pointing to pandemic social isolation as a trigger for the onset and worsening of these conditions [5,10].

Moreover, it is also essential to evaluate the effects of social isolation among different groups. Studies indicate that even though social isolation induces alterations in psychiatric conditions for many people, some groups, such as men and women, can present slightly different responses. It is important to understand the social and biological gender peculiarities, making it possible to elucidate the different psychiatric manifestations demonstrated [11,12].

Therefore, multiple facets of this imposed isolation and its consequences to global health still need to be evaluated in order to provide a better understanding of the phenomena experienced during COVID-19 social isolation and their persisting effects.

## 2. Methods

A narrative review study was developed to analyze the context of psychiatric disorders due to social isolation stress during the COVID-19 pandemic. Initially, comprehensive research was undertaken to evaluate which dysfunctions were most affected by social isolation and more prevalent in the scientific literature to determine the aspects of mental disorders to be approached. In this sense, the conditions included autism spectrum disorder (ASD), learning disabilities, schizophrenia, dementia, depression and anxiety, bipolar disorder (BD), eating disorders, and substance abuse.

After that, the reviewers sought research articles in the primary databases related to (1) social isolation; (2) social isolation stress; (3) the COVID-19 pandemic; and (4) the identified diseases in order to collect the most important data concerning the impacts of social isolation stress on those disorders. Therefore, the articles included in this review were the publications found to be most relevant and related to the purpose of this study, i.e., those evaluating the mental disorders with respect to groups with different age ranges and genders.

Finally, the reviewers analyzed the collected data related to the influence of social isolation stress in the context of each disorder and the possible measures to be developed, intending to offer a complete review of the impacts of the isolation period, as well as the potential management strategies emerging in this scenario. 

## 3. Impact of Social Isolation on Neuropsychiatric Disorders

### 3.1. Autism Spectrum Disorder

Autistic spectrum disorder (ASD) was first described in 1942 as an alteration in the individual’s neurodevelopment. Characteristics of ASD include deficits in communication and social interaction and restricted and repetitive patterns of behavior and interests [13]. The difficulty in adapting to changes, such as routine interruption, associated with a lack of individual support, exacerbates the negative behaviors present in ASD [14].

Social isolation is a characteristic of individuals with ASD, and several studies and treatments aim to improve this characteristic. Despite the importance of controlling the social isolation of individuals with ASD during the COVID-19 pandemic, isolation was inevitable. The damage caused directly affected the mental health of the population in general, but it had a greater impact on individuals with ASD [15].

An observational study described the relationship between the genetic risk of social isolation and its impact on various mental health-related outcomes, including ASD. Although not well characterized, studies suggest the existence of genetic risks of social isolation associated with an increased risk of developing autism and other neuropsychiatric disorders such as depression and psychosis. The authors suggest that genetics may play an essential role in developing certain behavioral and psychological traits, including a propensity for social isolation, and that these traits may increase the risk of developing disorders, including autism [10]. It is therefore possible to consider that individuals with ASD are even more likely to develop negative pathological symptoms in moments of established social isolation.

Breaking the pre-pandemic routine was a relevant harmful factor, both for individuals with ASD and for their family members. When schools moved their classes to the remote format, families of individuals with ASD found it more difficult or unfeasible to organize a new routine that could have a less negative impact on family life. Therefore, anxiety levels were higher in children with ASD and their families than those without it [13].

Several clinical studies targeting ASD needed to be adapted during the pandemic. Clinical trials had to be stopped due to COVID-19. Some clinics temporarily closed, suspending research and face-to-face behavioral intervention activities such as parenting therapy and training. Subsequently, experimental protocols were adapted to enable their continuity [16].

### 3.2. Learning Disabilities

According to the APA-5 (American Psychiatric Association, 2013—Diagnostic and Statistical Manual of Mental Disorders, 5th Edition Washington, DC, USA), “learning disabilities” are a broad category of disorders that can affect a person’s ability to learn or use specific academic, social, or practical skills. Individuals with these conditions may struggle with reading, writing, math, or communication. These difficulties are not caused by intellectual disabilities, lack of motivation, or poor teaching; they persist over time [17].

Such conditions can significantly impact an individual’s life, as it increases the propensity for low academic performance, mental health problems, difficulties in the job market, and lower quality of life [18]. The risk of developing learning disabilities may be related to genetic predisposition and environmental factors, such as poverty and exposure to toxins, thus affecting brain development and culminating in a learning disability diagnosis [19].

There are multiple forms of learning disabilities, such as dyslexia, dysgraphia, dyscalculia, and auditory processing disorder [20]. Each of these disabilities affects a specific area of learning and can manifest itself in different ways. Therefore, it is essential to understand the various manifestations of each as symptoms can vary widely from one individual to another [17].

Social isolation can harm the academic performance of students with learning disabilities as it affects social issues related to feelings of loneliness, such as peer rejection and distancing. When socially isolated, these students may miss opportunities to learn from peers, practice social skills, and develop positive self-esteem. Furthermore, social isolation can lead to a lack of motivation and learning engagement, negatively affecting academic performance [21].

For students with a learning disability, online classes and social isolation can result in fewer opportunities to practice social skills and develop meaningful relationships with peers. Furthermore, the sudden shift to online learning can create additional challenges, such as a need for access to assistive resources and technologies, teacher interaction, and direct support [21].

Students or not, all individuals with learning disabilities are negatively affected by social isolation. One study observed that the COVID-19 pandemic and social distancing measures resulted in significant changes in the daily lives of adults with learning disabilities, including the interruption of routine activities and a lack of social interaction with family, friends, and health professionals. These measures impacted their mental and physical health and their ability to carry out daily activities and work. The study evaluated occupational therapy referrals received before and during the height of the COVID-19 pandemic and found an increase in referrals related to anxiety, depression, and social skills during the pandemic, emphasizing the importance of occupational therapy for these adults [22].

Although they are still few, studies have shown that the interruption of routine activities and the lack of social interaction with friends and health professionals have an impact on the mental and physical health of individuals with learning disabilities, as well as on their ability to carry out daily activities and work.

### 3.3. Schizophrenia

Schizophrenia is a complex and chronic mental disorder that affects approximately 1% of the global population. It is characterized by a combination of positive symptoms, such as hallucinations and delusions, negative symptoms, such as reduced emotional expression and motivation, and cognitive symptoms, such as concentration and memory difficulties [23].

Social isolation can contribute to worsening symptoms and quality of life in individuals with schizophrenia. Individuals who experienced social isolation during the pandemic had significantly higher psychological distress levels than the control group. More specifically, individuals with schizophrenia reported higher levels of anxiety, depression, and stress symptoms and lower levels of subjective well-being and social support. These results suggest that the social isolation measures implemented during the COVID-19 pandemic harmed the psychological well-being of hospitalized patients with schizophrenia. The lack of social interactions and reduced access to support systems may have contributed to these individuals’ increased psychological distress [24].

Moreover, it is important to mention that these individuals, considered a subgroup with a higher risk of mortality if infected by SARS-CoV-2, had an especially rigorous recommendation for social isolation despite the effects it might have had on their psychiatric conditions. In this sense, studies indicated that—more than the social isolation itself—the spread of misinformation related to the virus affected the mental health of schizophrenic patients in isolation, contributing even more to their mental instability and the development of psychiatric symptoms, along with the development of other mental health problems, such as the exacerbation of symptoms related to depression, anxiety, and stress [12]. Therefore, the impact of social isolation and the pandemic was severe and harmful for these patients.

In another study, researchers found that individuals with schizophrenia who experiences suicide ideations presented a six-fold increase in suicide. In contrast, the relationship between those in patients with mood disorders was found to be statistically irrelevant [25]. Thus, even when compared to other psychiatric conditions, schizophrenic patients were more impacted by social isolation, leading to these concerning data related to their levels of suicidal ideation and suicide attempts.

Furthermore, De Donatis et al. (2022) studied the effects of social withdrawal on the neurocognition of schizophrenic patients. Their research found that social isolation had critical adverse effects on neurocognition, affecting parameters such as verbal memory, processing speed, and working memory domains, therefore impacting the functionality of these individuals [26]. Considering what has been revealed about the exacerbation of schizophrenic symptoms, the development of other psychiatric dysfunctions, and the worsening in functionality, it can be concluded that schizophrenic patients suffered a global impact on their mental health due to social isolation. 

### 3.4. Dementia

Dementia is a neurodegenerative condition that mainly affects memory, thinking, behavior, and the ability to carry out daily activities, with Alzheimer’s being its most common form. Social isolation is a common phenomenon among people with dementia, and it can occur due to several factors, such as stigma, communication difficulties, and physical limitations. The relationship between dementia and social isolation is complex and can significantly negatively affect the health and well-being of those affected [27].

Studies have shown that social isolation is associated with a higher risk of developing and progressing dementia. A meta-analysis of longitudinal studies showed that social isolation is associated with a 50% increase in the risk of dementia in the elderly [28]. Additionally, loneliness and social isolation have been linked to faster cognitive decline in people diagnosed with dementia [29].

Social isolation can also negatively impact the mental health of people with dementia in a manner unlike other conditions already mentioned. A study by Victor et al. (2012) showed that social isolation is associated with a higher risk of depression in these individuals [30]. Loneliness and isolation can also lead to symptoms of anxiety and stress, worsening the quality of life of people with dementia [31].

In addition to increasing the behavioral symptoms of psychiatric disorders, social isolation and a lack of support for individuals diagnosed with dementia can cause reduced cognitive functionality. A qualitative study evaluated the experiences and perceptions of people with dementia and their caregivers regarding the closure of social support services during the COVID-19 pandemic, identifying the significant negative impact of these services on the quality of life and well-being—in particular, the emotional well-being—both patients and caregivers [32].

Individuals with dementia may have atypical clinical symptoms of infectious diseases such as COVID-19, in which subtle or less specific symptoms can be identified, making accurate diagnosis difficult. Of note is the importance of considering underlying cognitive impairment when assessing and monitoring patients with dementia during unusual periods [33].

Systematized studies are still needed in order to assess the impact of social isolation on individuals with dementia. However, the scientific literature already includes some results pointing to a greater risk of disease development and progression, faster cognitive decline, mental health problems, and caregiver burden associated with measures undertaken during the pandemic.

### 3.5. Depression and Anxiety

Depression and anxiety are prevalent mood disorders worldwide. Their etiology and pathophysiology are complex and multifactorial. A persistent state of sadness, hopelessness, and loss of interest or pleasure in day-to-day activities characterizes depression. These symptoms cause significant impairment to social, occupational, or other vital areas of the person’s life. Anxiety is characterized by excessive worry or persistent fear, disproportionate to the circumstances, which interferes with the person’s normal functioning. Anxiety can be classified in many ways, such as generalized anxiety disorder, panic disorder, and post-traumatic stress disorder [20].

Stressful situations contribute to the triggering or worsening of these disorders. The COVID-19 pandemic created a favorable environment based on social isolation and fear of the unknown. Diagnoses of depression and anxiety have increased significantly in recent years [34]. According to WHO data, there was an increase of at least 25% in cases of depression and anxiety in the first year of the pandemic alone, which may have been even more significant in the following years [35].

During the second wave of the pandemic, a study in Myanmar identified 38.7% of patients infected with COVID-19 as exhibiting depressive symptoms, and the significantly associated factors were age equal to or greater than 40 years, family with fewer than four people, low family income, and infection in family members [36]. In another study, low serum 25-hydroxyvitamin D levels and a diagnosis of depression were identified as predictors of greater vulnerability to the stressful impact of the COVID-19 outbreak [37].

Levels of anxiety and depression increased in patients with myasthenia gravis during the COVID-19 pandemic and quarantine period in India. Furthermore, the negative impacts on mental health was more pronounced in patients with greater disease severity [38]. Individuals placed in quarantine centers due to the pandemic in Nepal also had increased levels of anxiety and depression. Such conditions were related to the fear, loneliness, and frustration associated with quarantine [39].

The comorbidity between depression and insomnia has become more prevalent, mainly affecting patients who contract COVID-19. Combining these two disorders can lead to a negative cycle, where depression can exacerbate insomnia and vice versa. In addition, quarantine and social isolation have contributed to increased emotional stress and sleep disturbances, further aggravating the relationship between depression and insomnia. In patients positive for COVID-19, this comorbidity may be even more pronounced due to the physical and psychological impacts of the disease. Uncertainty surrounding recovery, health concerns, and the possibility of complications can intensify depressive symptoms and difficulty sleeping [40].

Most research revealed a significant increase in the number of diagnoses and a worsening of anxiety and depression symptoms, both due to the stress caused by the psychosocial situation of the pandemic and by SARS-CoV-2 infection [34]. In this sense, research suggests that this stress was caused mainly by feelings of loneliness in this period, doubts related to the moment of global insecurity, and fear of being severely infected, culminating in negative effects on mental health [34]. 

Therefore, social isolation and the global pandemic scenario could induce a rise in the number of individuals suffering from anxiety and depression, conditions which are already very prevalent worldwide [20,34]. This reality generates the need for careful management of these critical and incident conditions (related to already-worrying psychiatric disorders) by health professionals in the years following this phenomenon.

### 3.6. Bipolar Disorder

Bipolar disorder (BD) is a chronic condition with a high risk of suicide and characterized by alternating episodes of (hypo)mania and depression intertwined with euthymic phases, relatively free of symptoms. BD can be subdivided into BD type I and type II, and what differs between each category is that in BD type II, the individual experiences depressive and hypomanic episodes without a complete manic episode [41]. 

In general, social isolation increases the probability of mortality when associated with BD. One study identified a significant effect of social isolation on the chances of mortality, with the increased probability of death being 26% for reported loneliness, 29% for social isolation, and 32% for living alone. Some indicators of social isolation involve living alone, having few social network ties, and having infrequent social contact, conditions usually present in BD’s clinical symptomatology [42].

During social isolation, people diagnosed with BD may face additional challenges. A lack of social interaction and emotional support can lead to feelings of loneliness, isolation, and worsening depressive symptoms. A lack of routine and social structure can also negatively affect the course of the disorder. Furthermore, social isolation can reduce social and leisure activities, interfering with the quality of life and emotional well-being of individuals with BD. The lack of social interaction can also make identifying and managing BD symptoms challenging and increase the risk of relapse [43].

During the quarantine period, a 32-year-old woman with no previous psychiatric diagnosis had an acute manic episode during a stressful situation at the beginning of the pandemic. Symptoms included elevated mood, increased energy, fast speech, grandiose thoughts, and impulsive behaviors [44].

Few studies still relate BD to the effects caused by social isolation and quarantine due to COVID-19. Despite this, the published research corroborates that the COVID-19 pandemic and associated social isolation measures can also predispose or intensify BD cases. Physical isolation, social distancing, and social interaction restrictions can lead to a greater sense of loneliness, increased stress, and decreased social support, which can negatively affect the mental health of these patients. In addition, limitations in accessing mental health services due to the pandemic may also contribute to greater stigmatization and difficulties in inadequate treatment.

### 3.7. Eating Disorders

Eating disorders are psychiatric conditions characterized by eating behavior, body image, and weight perception disturbances. These conditions may include anorexia nervosa, bulimia nervosa, binge eating disorder, and other unspecified eating disorders [45]. They are considered complex conditions resulting from a vast interaction of genetic, neurobiological, psychological, and social factors affecting individuals worldwide, which are diagnosed mainly in adulthood [46].

The pandemic has significantly impacted eating disorders, with reports of increases in both symptoms and risk for these disorders. Major concerns include changing eating patterns, increasing restrictive behavior, and increasing concern about weight and body shape. Social isolation associated with pandemic-related stress and the disruption of mental health services contributes to worsening symptoms [47].

In a study conducted in Australia, participants who reported a history of eating disorders indicated an increase in dietary restriction and the number of binge eating episodes. In contrast, the general population reported increased consumption of unhealthy foods and reduced physical activity. These changes can be attributed to several factors, including the availability of specific foods, as well as increased stress, anxiety, and depressive symptoms caused by social distancing measures. These results suggest that the pandemic has significantly impacted eating and exercise behaviors in individuals with eating disorders and the general population [48].

Another study of people in the United States and the Netherlands showed that the pandemic significantly impacted individuals with eating disorders. About 62% of participants reported that the pandemic had worsened their symptoms, while just 8% reported an improvement. The most-affected symptoms were a preoccupation with weight, body shape, and food restrictions. Additionally, most participants reported changes in eating behaviors during the pandemic; about 30% of participants reported an increase in food restriction, 21% reported an increase in binge eating episodes, and 28% reported a rise in exercise-related preoccupation. Corroborating with the other studies discussed, the factors contributing to these negative changes included social isolation, fear of gaining weight during confinement, and the interruption of face-to-face treatments and therapies [49].

A survey carried out in Germany noted that most individuals diagnosed with anorexia nervosa reported a worsening of symptoms during the pandemic. About 62% of participants reported an increase in concern about weight and body shape, and 36% reported an increase in dietary restriction. Additionally, 39% of participants reported an increase in concern about exercise. Contributing factors to these negative shifts included social isolation, discontinuation of face-to-face treatments and therapies, fear of gaining weight during lockdown, and limited access to proper food and exercise [50].

A randomized controlled clinical trial with two groups, one that was exposed to thoughts related to social distancing and a control that was exposed to neutral ideas, found that thoughts related to the experience of social distancing had a significant impact on dietary intake and the hypothesized trend. For episodes of binge eating, participants exposed to thoughts of social distancing reported increased food intake and a greater propensity for binge eating episodes compared to the control group. These results suggest that the experience of social distancing during the COVID-19 pandemic may have triggered changes in eating behavior, leading to greater food consumption and a greater tendency towards binge eating episodes [51].

Female college students with body dissatisfaction and risk of developing eating disorders experienced a negative effect on body perception and body-related behaviors during the COVID-19 pandemic. There has been increased body dissatisfaction, concern about body weight and shape, and compensatory practices such as dietary restriction and excessive exercise by these women during the pandemic [52].

Individuals with a previous diagnosis of binge eating disorder have also been affected by the COVID-19 pandemic. Research conducted during the first lockdown in Germany revealed an increase in the frequency and severity of binge eating episodes and an increase in compensatory behaviors such as food restriction and excessive exercise. Furthermore, these individuals reported increased symptoms of depression and anxiety during the pandemic, possibly related to increased disordered eating behaviors [53] (Figure 1).

### 3.8. Substance Abuse

Before the COVID-19 pandemic, substance abuse already represented a serious issue to public health systems worldwide, with the number of affected individuals only increasing. Adolescents represented the age group at higher risk of the development of this condition due to their propensity to the experimentation of alcohol and drugs [54,55]. In the context of SARS-CoV-2, individuals with addiction behaviors were also significantly affected. They could be analyzed differently according to the socioeconomic segment to which they belonged. Individuals financially capable of maintaining social isolation were more likely to present worsening in their compulsions and consumption of drugs. However, they were at lower risk of infection. On the other hand, many people with substance abuse problems belong to marginalized segments of society, who were unable to be isolated and therefore under a higher risk of infection by SARS-CoV-2 [56]. In this sense, it is worth highlighting the need for studies related to the impacts of the pandemic on this subgroup as well.

Furthermore, in patients who suffer from substance abuse and could maintain social isolation, the whole protocol related to treating this condition was jeopardized. The main interventions and change of habits induced in the treatment of those patients are related to the stimuli of social interaction, family support, and the absence of the use of drugs [56]. Considering that social isolation does not allow socialization, a significant part of the therapy, i.e., that related to the psychological factors associated with a support network, could not be executed anymore, harming the mental health of those patients and worsening in their addiction behaviors [56].

Moreover, studies also observed that social isolation led to stressful conditions and alterations in mental health that contributed to the use of drugs by adolescents. Temple et al. (2022) found that social isolation induced high stress levels, feelings of loneliness, financial problems, and family issues, all related to an increase in the use of drugs by adolescents [57]. Moreover, it is essential to mention the impact of social isolation on alcohol abuse, especially since, according to researchers, there was a very significant increase in the consumption of ethanol during this period. Along with this, it was hypothesized by researchers that a significant association between family stress, job losses, anxiety related to the virus, and the increase in drinking habits might be responsible for the findings [58].

Finally, it is relevant to mention that elevation in the consumption of alcohol can imply an increase in aggressive behavior, domestic violence, and child abuse; therefore, it is an extremely dangerous dysfunction [58]. Thus, social isolation had a high impact on patients treated for substance abuse and also served as a trigger for people with tendencies towards it, representing a severe condition that can have dangerous consequences for their families and deserves further study. 

## 4. Social Isolation Impacts for Men and Women

Initially, both men and women felt the impacts of social isolation. However, some differences can be evaluated. According to some studies, women are more prone to developing feelings of loneliness. Nonetheless, researchers found a higher prevalence of mental health alterations in men [11,59]. Moreover, studies suggest that women in some situations, such as pregnant and postpartum women, women who suffered miscarrying episodes, women who were experiencing domestic violence, or women who were single parenting, presented a higher risk for developing psychological dysfunctions during social isolation [60]. Nevertheless, the incidences of psychiatric admissions during the COVID-19 pandemic were mainly composed of men [61].

Among the hypotheses related to the differences in women’s and men’s manifestations of mental health impairments, one of the main theories developed was the influence of having or not having a support network, i.e., the capacity to talk about feelings and problems with friends or family. Moreover, men are more resistant to seeking help from mental health professionals, presenting tendencies to search for professional help only in severe scenarios and to isolate themselves from other people when depressed or anxious [11,61].

When evaluating cognitive instead of psychological conditions, the pattern changes. In this scenario, women tend to suffer a worsening of their cognitive conditions when in social isolation, while men do not have their cognitive functions altered by the isolation. Furthermore, the absence of social support can also induce lower cognitive function and social isolation in women. Neither of the evaluated alterations impairs men’s cognitions [62].

Thus, different mental health and cognitive conditions can be affected by social isolation and induce different responses in men or women. Different factors can also influence this situation, with support networks representing a very relevant condition in the preservation of mental functions, as well as individual life factors, such as employment status or being a student [11,60]. Hence, further studies must be developed to elucidate the different factors that participate in the altered manifestations between genders, allowing for the management of men and women according to their particular characteristics, influences, and risks.

## 5. Strategies for Mitigating Neuropsychiatric Impairments

The COVID-19 pandemic has unleashed a series of global challenges that go beyond physical health issues. Social isolation measures, essential to containing the spread of the virus, have significantly impacted the mental health and quality of life of individuals worldwide, especially those who were already facing neurological and neuropsychiatric conditions. The restrictions imposed by social distancing, the interruption of daily routines, and the lack of social interactions have contributed to worsening the symptoms of these conditions, increasing the risk of emotional crises, and worsening quality of life. In this context, non-pharmacological therapeutic strategies emerge as promising approaches to mitigating the adverse effects of the pandemic and isolation, offering holistic support adapted to individual needs. This section explores studies on strategies used to minimize the neuropsychiatric damage arising from the pandemic and, consequently, to improve the quality of life of vulnerable individuals.

Regarding ASD, regular access to healthcare is essential to monitoring its development concerning social communication, perceptual–motor, and cognitive comorbidities. During the pandemic, health interventions through telehealth were necessary, as they allowed health professionals to effectively guide families in performing games and activities that could improve the child’s strength, resistance, executive functioning, and social skills via TEA. The family-centered telehealth approach has shown positive results and has allowed families to maintain or carry out training, monitoring the child’s development even during social isolation [14].

For children between 3 and 6 years of age diagnosed with ASD, a training intervention for parents based on the Developmental Intervention Program (DIR/Floortime) approach, carried out by psychologists and occupational therapists, was beneficial. The intervention ensured significant improvement in ASD symptoms regarding emotional, functional development, and adaptive behavior. Additionally, parents of children with autism who received the training intervention reported significantly reduced stress levels. Therefore, parent training intervention may be effective in improving autism symptoms and emotional and functional development in children with ASD and reducing parental stress [63].

The pandemic had unfavorable effects on people receiving mental health care, leading to the use of digital health tools. Because of this, the use of smartphones and social media has become increasingly common among individuals with neuropsychiatric disorders. Given this, exploratory research analyzed the impact of smartphones and social media on mental health, including their potential influence on relapse rates in schizophrenia, and found that social interaction could be a possible supporting therapeutic strategy in the context of limited resources [64], such as in the COVID-19 pandemic.

Another study found that smartphone-facilitated social activity may be an essential metric for determining the risk of relapse in individuals with schizophrenia. Monitoring digital data can provide access to sensitive, meaningful, and ecologically valid information previously unavailable in routine care. Researchers followed individuals with schizophrenia at high risk of relapse for a year who used smartphones with a behavioral detection system called CrossCheck. The study aimed to examine the relationships between social behavior and the occurrence of relapses. The research showed that reductions in the number and duration of calls made and in the number of text messages were associated with relapses [65].

Another study analyzed the effect of animal-assisted therapy, showing a positive impact on social interaction and quality of life in patients with schizophrenia during the COVID-19 pandemic. Participants reported improvements related to social interactions and more effective communication, in addition to experiencing a greater sense of satisfaction. In addition, quality of life in general, including the physical, psychological, social, and environmental domains, showed a significant improvement after involvement in therapeutic sessions [66].

Additional measures are needed in order to ensure the safety and well-being of people with dementia when social isolation is unavoidable. These include implementing infection prevention strategies such as proper hygiene, using personal protective equipment, restricting visits to long-term care facilities, promoting cognitive activities, practicing effective communication, and managing challenging behaviors. It is also necessary to ensure emotional and psychosocial support for patients and their caregivers, who may face social isolation, interruption of support services, and increased stress [67].

In adolescents, an intervention that included educational materials and activities focused on improving resilience and well-being significantly protected stress levels and enhanced physiological outcomes, reducing cortisol, heart rate, blood pressure, and psychological consequences, demonstrating improved self-perception and a better ability to deal with stressful situations [68].

The mindfulness-based mobile health (mHealth) intervention, performed using a mobile application with a convenient and flexible platform, also effectively reduced psychological distress and improved well-being. Participants in the intervention group demonstrated increased mindfulness, self-compassion, and psychological resilience skills. The randomized clinical trial was conducted on university students between March and April 2020, the critical quarantine period for COVID-19 [69]. Another study with undergraduate students undertaken during the quarantine period showed that higher levels of social support were associated with lower levels of depression, anxiety, and stress [70].

Video sessions for cognitive–behavioral intervention in university students with severe COVID-19 anxiety reduced anxiety, suggesting once again that technology-mediated interventions are beneficial in lowering stress symptoms caused by the pandemic [71]. Similarly, another study on isolated individuals during the first lockdown in Italy found that online psychological counseling significantly reduced anxiety symptoms and adverse effects, increased well-being, and decreased psychological distress [72].

Corroborating the above studies, a randomized clinical trial involving individuals diagnosed with COVID-19 revealed that an integrated intervention associating cognitive–behavioral therapy and progressive muscle relaxation sessions significantly improved immunological biomarkers. Individuals who received the intervention also showed a reduction in the severity of COVID-19 and slower progression of the disease compared to the control group, as well as a decrease in stress, anxiety, and depression [73].

Psychoeducation is a therapeutic strategy used in individuals diagnosed with BD and their families. One study observed that psychoeducation positively impacted family members’ attitudes toward psychological disorders. The group that received psychoeducation sessions showed significantly improved knowledge and understanding of BD and reduced levels of internalized stigma. This study highlights the importance of involving family members in the treatment and support process, as their attitudes and beliefs can significantly impact the well-being and recovery of individuals with BD [74].

Specific inhibitory control training for food, a technique that aims to improve people’s ability to control impulses, make more rational decisions, and avoid impulsive behaviors, through a mobile application, demonstrated positive effects in reducing perceived hunger and liking for energy-dense foods, as well as on symptoms of depression in people with disinhibited eating behaviors. However, no significant effects were observed in reducing binge eating symptoms. The results of this randomized clinical trial suggest that food-specific inhibitory control training may have a limited impact on binge eating. Combining this approach with other types of exercise may be necessary in order to achieve better results in this area [75].

A reward retraining protocol consisted of cognitive–behavioral training to help participants strive for healthy rewards and reduce their cravings for unhealthy foods, effectively reducing binge eating episodes and improving the quality of life of individuals who suffered from binge eating during the COVID-19 pandemic [76].

The COVID-19 pandemic has brought unique challenges for individuals already struggling with neuropsychiatric disorders. However, non-pharmacological therapeutic strategies have emerged as a fundamental approach to improving quality of life and reducing social isolation’s adverse effects. By implementing interventions such as teletherapy, self-care practices, adapted physical activity, relaxation techniques, and the promotion of virtual social interactions, it is possible to provide comprehensive support that addresses both the physical and emotional needs of these individuals. While pharmacological solutions play a crucial role, non-pharmacological approaches can empower patients to face today’s challenges more resiliently and positively (Figure 2).

## 6. Discussion

The COVID-19 pandemic has had many impacts on global health, with social isolation being responsible for the development of mental health impairment in a surprisingly large number of individuals [2]. In this sense, the psychological effects caused by the stress of isolation were relevant, demonstrating the importance of understanding the mechanisms involved in these processes. This study sought to centralize the main effects and their possible causes; however, further studies are still needed in order to deepen our understanding of the stress caused by biopsychosocial phenomena associated with the COVID-19 pandemic.

Even with our lack of understanding of the biological and psychological mechanisms impacted by social isolation, the influence of the shared environment on population health is evident. Thus, modifying social habits—in addition to a break in habits and routine for some individuals—represents a contradiction of human nature, which requires, at the level of the community, interaction and social coexistence in order to achieve optimal health conditions [1,18,37]. In this way, one can understand the origin of the wide incidence of disorders emerging during the period of isolation and the importance of this type of analysis for both individual and collective health.

It is essential to mention the relevance of this type of study and relate it to individual psychological issues that generate self-imposed isolation as well, i.e., not only the isolation caused by the pandemic, but also Internet addiction, for example, which can cause (or be caused by) social isolation and impact depressive conditions [77]. This self-isolation reality is very present today, often due to the pandemic, and it can adversely affect individuals’ physical and mental health.

Data from the scientific literature concerning studies carried out during and immediately after the period of social restriction imposed by the COVID-19 pandemic present much evidence that stress has a relevant impact on the mental health of individuals [1]. The stress generated by fear of the disease and social uncertainties increased the symptoms and suffering of people with neuropsychiatric disorders, and it precipitated the onset of disorders in individuals with some biopsychosocial vulnerability [78].

However, even with the scientific community racing to produce research that could reveal the impact of the pandemic on health, there is still time to analyze the established consequences that the stress generated may trigger. In this sense, post-COVID-19 studies, which are on the rise globally, have the potential to reveal significant results that support the possibilities of therapeutic intervention.

On the other hand, the results regarding intervention strategies during the pandemic reveal the beneficial effects of specific behaviors on reducing the damage caused by isolation. This strategy’s results may lead to future protocols to be implemented and research protocols that can be associated with other interventions, enabling a breakthrough in the production of therapeutic intervention strategies to alleviate the damage caused by the pandemic.

It is relevant to mention the importance of adaptations to digital tools and of other technologies implemented during this period to make mental health care possible in isolation; the demand for this care practice model has many potential applications and can also be used in other contexts related to the pandemic, as its benefits have proven to be relevant [10,48,55].

Finally, studies that gather data on specific mental health conditions help professionals build a unique therapeutic plan by which to qualify their assistance, aiming at still-more-excellent treatment outcomes. Thus, the conclusions of these analyses can be instituted in the context of the clinic, offering better handling and precision in patient care.

## 7. Limitations and Strengths

This study’s main limitations pertain to the evaluation of different article designs—with no age range filter, for example—in comparing types of studies and individual realities that different factors can influence. Moreover, not all disorders could be approached similarly since some were not as well-represented in the literature as others; hence, our analysis could not be developed at the same depth with respect to all the chosen dysfunctions. Therefore, the most relevant limitation of the study is its need for more homogeneity in the data evaluated.

On the other hand, as the main strengths of this study, we can mention the importance of comprehensive analysis, our evaluating the main neuropsychiatric disorders affected by social isolation stress, and our considering data from different countries, age ranges, genders, and possible management strategies. Moreover, our summarization of these heterogeneous types of data represents a rich material, since it can offer a variety of information about the possible impacts of social isolation, the manifestations of those impacts in different contexts, and how distinct populations are being treated. 

Furthermore, the importance of the section evaluating the possible measures to reduce neuropsychiatric impairment due to the social isolation period associated with the pandemic must be highlighted. This section is of particular interest since it summarizes the primary management strategies among the articles analyzed, offering potential new therapeutics by which to improve the conditions of patients suffering from the influence of the isolation scenario. 

## 8. Conclusions

The COVID-19 pandemic has impacted many types of dysfunctions through the imposition of social isolation, especially psychiatric conditions. Patients living with those disorders presented with worsening symptoms, demonstrating the importance of social interaction. Moreover, social isolation alters individuals’ habits and routines, affecting their psychological states. Therefore, this experience represents the importance of relationships between people and the need for interaction in maintaining human health.

Furthermore, to allow health professionals to improve the management of patients who suffer from worsening psychiatric conditions due to social isolation, it is vital to understand the relevance of the situation imposed by the pandemic and the impact it represents for patients. Along with this, the individualization of treatments and the analysis of the disease suffered by each patient can help professionals to recognize which change caused by social isolation represents the worst influence on the individual, and thus the measure to be taken. In this sense, many possible measures were presented in the study. However, more interventions must be evaluated in order to improve the treatments offered by health services. 

## Figures and Tables

**Figure 1 brainsci-13-01414-f001:**
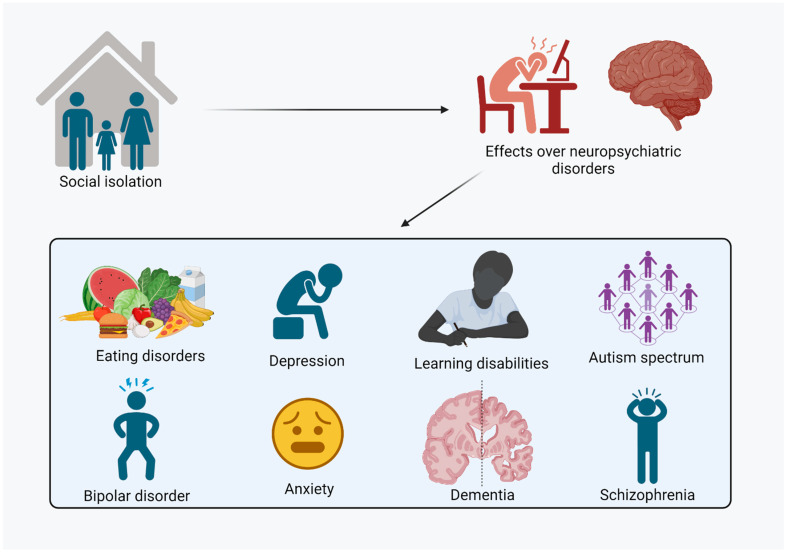
Main disorders affected by social isolation. Social isolation stress can lead to neurologic and psychiatric disorders. In this sense, among the main neuropsychiatric disorders triggered or modified by social isolation stress are learning disabilities, dementia, schizophrenia, autism spectrum disorder, eating disorders, anxiety, depression, and bipolar mood disorder.

**Figure 2 brainsci-13-01414-f002:**
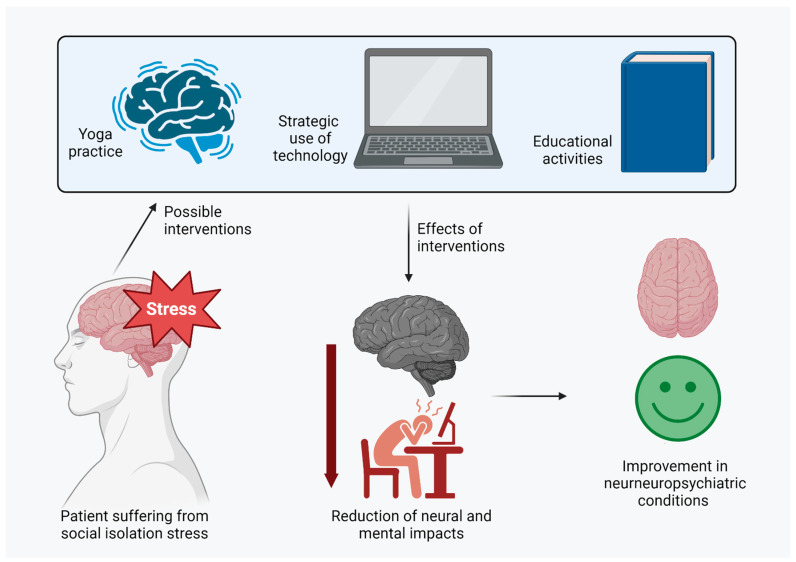
Possible therapeutic strategies against social isolation stress effects. To avoid the neuropsychiatric impact of social isolation stress, some of the measures that presented successful outcomes were the practice of mindfulness; the strategic use of technology, which allows health services to execute activities during isolation; and educational activities, i.e., explaining the nature of the disorders and some manners of dealing with them. As a result, these measures reduced the impacts of stress on individuals’ mental and neural status, improving neuropsychiatric conditions.

## Data Availability

The cited studies are publicly available.

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
