# Peer review of "The Impact of Stress from Social Isolation during the COVID-19 Pandemic on Psychiatric Disorders: An Analysis from the Scientific Literature"

_brainsci, 2023, doi:10.3390/brainsci13101414_

Round 1

Reviewer 1 Report

Comments and Suggestions for Authors

This is a review investigating the impact of stress from social isolation during the COVID-19 pandemic on neuropsychiatric disorders. The paper is well written and the topic is of interest; however, there no novelty in investigating these phenomena in a narrative review design. I recommend major revisions of the paper.

ABSTRACT.

1- The abstract is not reporting the design of the study (narrative review) and the results are not adequately reported. Please, expand them.

INTRODUCTION

1- I recommend to expand the introduction by including some references regarding the differences of the impact of the covid-19 pandemic in mental health in different countries, and by gender.

METHODS

A few words about the methods used in the narrative review are needed. Inclusion/exclusion criteria, why where the mentioned disorders included, and why not others.

Are adults, and child populations included for all the disorders analysed? 

Gender differences are needed to be discussed. Do women and men differ in terms of mental health problems during the covid-19 pandemic? 

Is there any gender difference in patients suffering from a relapse or exacerbation of mental disorders, or those suffering from an onset of illness?

CONSIDERATIONS AND CONCLUSIONS

I recommend to add references in that section. It should be better renamed as Discussion. 

Conclusions should be moved to a new section.

Please, consider a subsection called "limitations and strenghts". Is there any future directions recommendations in that field?

Author Response

Dear Reviewer,

We appreciate your valuable review time and are grateful for your relevant criticisms and suggestions.

All corrections and additions are highlighted in yellow in the manuscript.

Here are our answers:

We added information in the abstract mentioning the type of review and the results. Also, we have expanded the “Introduction” by mentioning the effects of social isolation in different countries and genders. As suggested, we have posteriorly added a section called “Social Isolation Impacts for Men and Women” in the study. Moreover, we have created a section called “Methods”, to explain more about the study's methodology. Additionally, we have changed the name of the section previously called “Considerations and Conclusions” to “Discussion”, and have included references in the analysis made in this module. Furthermore, we have created two new sections called “Limitations and Strengths” and “Conclusion”.

Reviewer 2 Report

Comments and Suggestions for Authors

Respected authors, as this is an overview of psychiatric disorders (not one neurological was listed) I would omit neuro from the title. Additionally, some topics (disorders) were more represented than others (schizophrenia the least, while depression and anxiety were far less represented than let's say learning disabilities and eating disorders), while some were not listed (alcohol, drugs...) This should be addressed as schizophrenia is the core psychiatric diagnosis. Furthermore, there is no mention of addictions, no alcohol use disorder, no drugs etc., which is strange as COVID-19 most definitely impacted the drinking rates and so on.

An additional point, mentioning yoga while not the concept of spirituality (religion), meditation or even prayer is biased.

Author Response

Dear Reviewer

We appreciate your valuable review time and are grateful for your relevant criticisms and suggestions.

All corrections and additions are highlighted in yellow in the manuscript.

Here are our answers:

Response: We were thankful for the comments and considerations. We have improved the subtopic “Schizophrenia”, as requested, and tried to maintain a similar representation of the conditions analyzed. We have also included a subtopic, "Substance Abuse," which analyzes the use of alcohol and drugs during social isolation, as suggested. We have altered the mistaken term “yoga” to “mindfulness”, which is not a concept of spirituality, but cognition, avoiding bias, as mentioned. We kept the word neuropsychiatric because the topics include dementia and autism spectrum disorder, which are more suited to neurological changes with psychiatric disorders.

Reviewer 3 Report

Comments and Suggestions for Authors

Thank you very much for the opportunity to review this manuscript. Very interesting work, needed in the context of analyzing the effects of the covid-19 pandemic and strategies for dealing with them. My comment concerns only one fragment of the text with a footnote [37], it is not in English. Please correct it. I congratulate the authors on an interesting analysis.

Author Response

Dear Reviewer,

Response: We were grateful for the consideration and timely observation.

We corrected the paragraph.

All corrections and additions are highlighted in yellow.

Round 2

Reviewer 1 Report

Comments and Suggestions for Authors

The authors have substantially improved the paper, and followed the recommendations of the reviewers.

The abstract section has been improved, and the introduction have been expanded.

They have included a methods section for this narrative review, what makes it clearier.

The subtopic of schizophrenia have been expanded and adequately referenced.The authors renamed the discussion section and included limitations and strenghts data.

Author Response

We are thankful for the comments and considerations.

Reviewer 2 Report

Comments and Suggestions for Authors

Respected authors, you have addressed most of the raised points accordingly. The only remaining issue for me is the title, as both autism and dementia are psychiatric diagnoses (ICD clearly states so). The fact that neurologists also see and treat some of these patients has little to do with the topic of the article and mental health. In other words, calling something neuropsychiatric doesn't increase or decrease its value, but still strays away the fact that those are psychiatric disorders.

Author Response

We are grateful for the considerations. We have changed the title as suggested.
